# Accurate prediction of gene deletion phenotypes with Flux Cone Learning

Charlotte Merzbacher[1], Oisin Mac Aodha[1] & Diego A. Oyarzún [1,2] ✉

Understanding the impact of gene deletions is crucial for biological discovery, biomedicine, and biotechnology. Due to the complexity of genome-wide deletion screens, there is growing interest in computational methods that leverage existing screening data for predictive modeling. Here, we present Flux Cone Learning, a general framework designed to predict the effects of metabolic gene deletions on cellular phenotypes. Using Monte Carlo sampling and supervised learning, our approach identifies correlations between the geometry of the metabolic space and experimental fitness scores from deletion screens. Flux Cone Learning delivers best-in-class accuracy for prediction of metabolic gene essentiality in organisms of varied complexity (*Escherichia coli*, *Saccharomyces cerevisiae*, Chinese Hamster Ovary cells), outperforming the gold standard predictions of Flux Balance Analysis. We demonstrate the versatility of our approach by training a predictor of small molecule production using data from a large deletion screen. Flux Cone Learning can be applied to many organisms and phenotypes, without the need to encode cellular objectives as an optimization task. Our work offers a broadly applicable tool for phenotypic prediction and lays the groundwork for building metabolic foundation models across the kingdom of life.

Gene deletions impact cellular phenotypes in multiple ways, affecting how cells function, proliferate, and interact with their environment. Understanding the effect of such deletions is fundamental for basic biological discovery and a variety of applications. For example, identifying lethal deletions is key for developing new cancer therapies[1] or antimicrobial treatments that avoid drug resistance[2]. In biotechnology, nonlethal deletions are a powerful strategy to redirect chemical flux toward production of high-value compounds for the food, energy, and pharmaceutical sectors, using genetically engineered cells as an alternative to petrochemicals[3]. Thanks to progress in high-throughput technologies such as RNAi or CRISPR-Cas9[4–6], genome-wide deletion screens have revealed foundational insights in numerous domains, including the genetic basis of disease[7,8], drug target discovery[9], and genetic engineering[10]. Due to the cost and complexity of deletion screens, computational methods hold substantial promise to complement experimental approaches, for example by filling gaps in coverage, extrapolating predictions to new variants or conditions, or aiding experimental design.

In the case of metabolic genes, the gold standard is Flux Balance Analysis (FBA), a computational method that predicts metabolic phenotypes by combining genome-scale metabolic models[11] (GEM) with an optimality principle[12]. This technique can model many metabolic tasks, such as growth capabilities in various substrates[13], cell-specific auxotrophies[14], or responses to drug interventions[15]. FBA is particularly effective at predicting gene essentiality in microbes, i.e., whether a gene deletion leads to cell death. For various model microbes[16], FBA predicts metabolic gene essentiality with high accuracy, but its predictive power drops when applied to higher-order organisms where the optimality objective is unknown or nonexistent[17–19]. Other methods build on FBA to extend essentiality predictions to other relevant tasks; for example, gene Minimal Cut Sets[20] were developed to identify combinations of deletions that block specific cellular functions, with particular success for predicting synthetic lethal genes in cancer[21]. Alternative strategies for essentiality prediction include network-based methods[22] and sequence-based approaches that employ

[1]School of Informatics, University of Edinburgh, Edinburgh, UK. [2]School of Biological Sciences, University of Edinburgh, Edinburgh, UK.
✉e-mail: d.oyarzun@ed.ac.uk

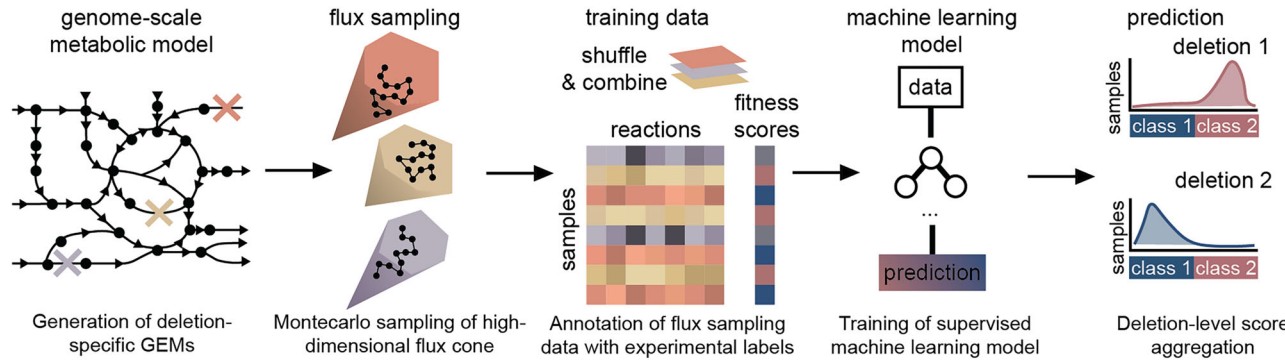

**Fig. 1 | Flux cone learning of metabolic deletion phenotypes.** The Flux Cone Learning pipeline predicts experimental fitness scores from changes to the shape of the metabolic space. Starting from a genome-scale metabolic model (GEM) of the wild-type, we first create deletion-specific GEMs, which are then sampled via a random walk defined in a high-dimensional cone. Flux samples from each cone are then stacked, randomly shuffled, and then paired with experimental fitness scores for supervised training. Predicted sample-level fitness scores are finally averaged across each deletion to produce gene-level predictions. The choices of supervised machine learning model and task (regression/classification) are flexible and can accommodate various types of deletion screens depending on the nature of the fitness readouts.

machine learning to extract predictive features from DNA or protein sequences[23–27].

Here, we describe Flux Cone Learning (FCL), a versatile machine learning strategy for predicting deletion phenotypes from the shape of the metabolic space. Through Monte Carlo sampling, our method utilizes mechanistic information encoded in a GEM to produce a large corpus of training data for each deletion. These data can be paired with experimental fitness readouts for a phenotype of interest and then employed for training predictive models with supervised learning. FCL can be adapted to multiple prediction tasks, provided that the fitness scores correlate with metabolic activity. This includes prediction of metabolic signals already encoded in a GEM, e.g., growth rate or the activity of specific pathways, as well as nonmetabolic readouts absent from the model but associated with metabolic activity. We show that FCL produces the most accurate predictions of metabolic gene essentiality, surpassing FBA predictions in all tested organisms. Crucially, FCL predictions do not require an optimality assumption and thus can be applied to a broader range of organisms than FBA. We demonstrate the flexibility of FCL for predicting other deletion phenotypes by building a predictor of small-molecule synthesis from deletion screen data.

## Results

### Learning the shape of the metabolic space

Our approach is based on learning the shape of the metabolic space of an organism through random sampling. FCL has four components (Fig. 1): a GEM, a Monte Carlo sampler to produce features for model training, a supervised learning algorithm trained on fitness data, and a score aggregation step. A GEM is defined by:

$$\mathbf{S}\mathbf{v} = 0, \tag{1}$$

$$V_i^{\min} \le v_i \le V_i^{\max}, \tag{2}$$

where $\mathbf{S}$ is an $m \times n$ integer matrix describing the metabolic stoichiometry, $\mathbf{v}$ is an $n$-dimensional vector of metabolic fluxes, and $(V_i^{\min}, V_i^{\max})$ are flux bounds that can be used to model gene deletions through a gene-protein-reaction (GPR) map. Upon deletion of gene $g_j$, the GPR determines which flux bounds need to be zeroed out in the GEM, i.e., by setting $V_i^{\min} = V_i^{max} = 0$ in Eq. (1); a single gene deletion can affect more than one reaction flux in the GEM. From a geometric standpoint, a GEM defines a convex polytope in a high-dimensional space, which is known as the flux cone of an organism[28].

The dimensionality of the cone equals that of the null space of $\mathbf{S}$, which for current GEMs can be up to several thousand dimensions depending on model complexity (Supplementary Fig. S1).

FCL relies on the observation that gene deletions perturb the shape of the flux cone, because zeroing out the flux bounds in Eq. (2) alters the boundaries of the polytope. The correlations between such geometric changes and a phenotype of interest can then be learned with supervised learning algorithms trained on experimental fitness scores. To test if the shape differences between flux cones can be captured from random samples, we first sampled five metabolically diverse pathogens (*Bordetella pertussis*, *Pseudomonas aeruginosa*, *Helicobacter pylori*, *Mycobacterium tuberculosis*, *Streptococcus pneumoniae*) from programatically generated GEMs to avoid confounders introduced by variations in model quality[29]. We trained a variational autoencoder[30] based on neural networks to compute low-dimensional representations of each species cone[31], using a large set of Monte Carlo samples of metabolic reactions shared across the five species and removing species-specific reactions. The learned representations are well separated across species, despite being trained on reactions shared by diverse species (Supplementary Fig. S2). This suggests that the cone geometry can be learned from Monte Carlo samples, and offers a path toward the construction of metabolic foundation models across many species and genomic perturbations.

To train predictive models of deletion phenotypes, FCL utilizes a Monte Carlo sampler to capture the shape of each deletion cone (Fig. 1). A supervised machine learning model is then trained on the flux samples alongside measured phenotypic fitness labels for each deletion; all samples in a deletion cone get assigned the same label. FCL does not prescribe the choice of machine learning model and can be applied to both regression and classification tasks. The feature matrix for model training has $k \times q$ rows and $n$ columns, where $k$ is the number of gene deletions, $q$ is the number of flux samples per deletion cone, and $n$ is the number of reactions in the GEM. This approach leads to large datasets; for example, in the case of the iML1515 model of *Escherichia coli*[13], acquiring 100 Monte Carlo samples for the 2712 reactions and 1502 gene deletions leads to a dataset over 3Gb in single-precision floating-point format. In the final step, FCL aggregates sample-wise predictions with a majority voting scheme to produce deletion-wise predictions.

### Best predictive accuracy of metabolic gene essentiality

We first tested FCL as a predictor of gene essentiality in *E. coli*, which has the best curated GEM in the literature. This evaluation allows

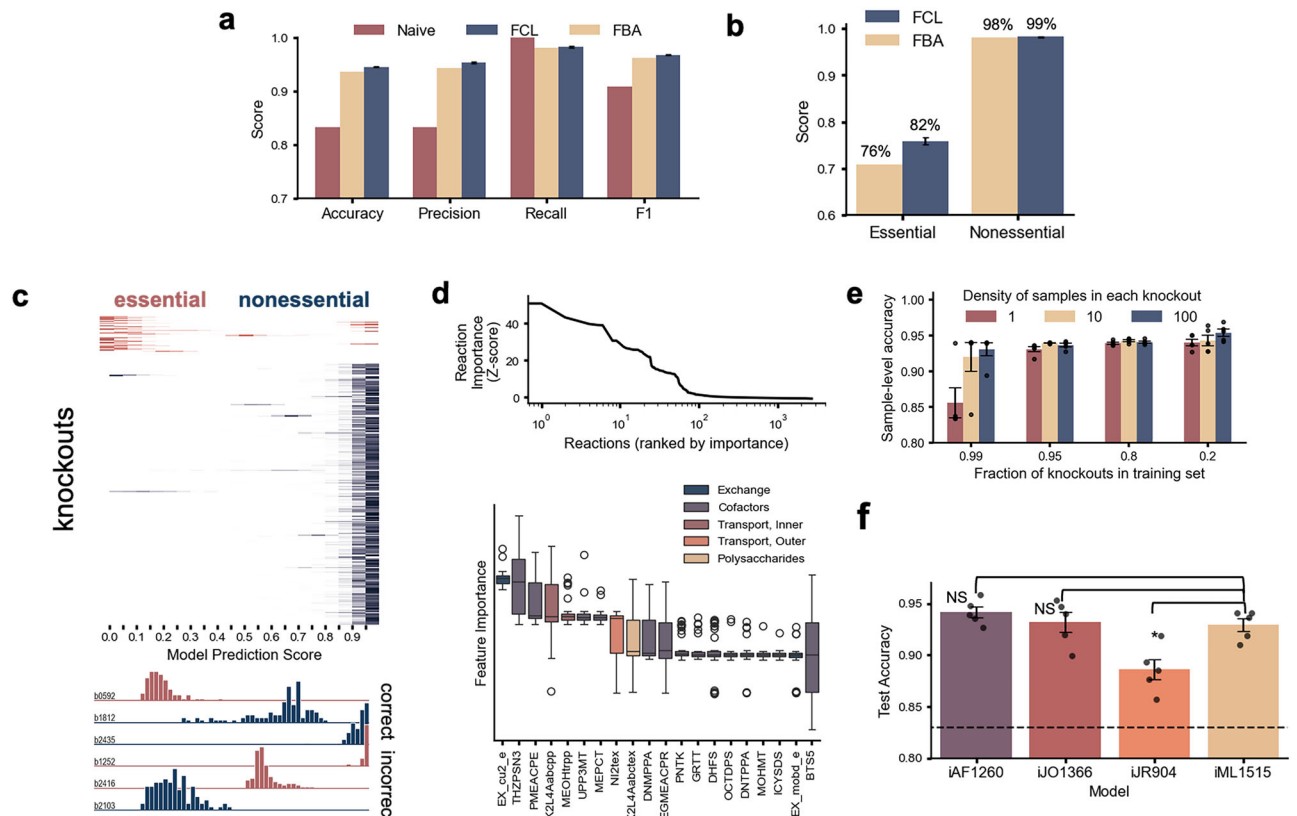

**Fig. 2 | Prediction of metabolic gene essentiality in *Escherichia coli*. a** Flux Cone Learning (FCL) delivers best results for metabolic gene essentiality prediction in *E. coli*, outperforming the current gold standard predictions from Flux Balance Analysis (FBA). FCL predictions were computed on a class-stratified test set with 300 genes held out from training; FBA predictions were computed for all genes in the iML1515 genome-scale model with default media with glucose as carbon source and aerobic growth[13]. The naive baseline was computed by predicting all genes as nonessential (majority class); training sets were subsampled from large flux sampling data with more than 5000 samples/cone. Receiver operating characteristic (ROC) curves for FCL and FBA predictions are shown in Supplementary Fig. S5. **b** Prediction accuracy for essential and nonessential genes. In (**a**, **b**) error bars denote mean ± standard error computed across 5 training repeats and 10 training subsamples each with $q = 100$ samples/cone (total $N = 50$ repeats per bar). **c** Top: Distribution of sample-level FCL prediction scores across 300 test genes for one representative random forest model; for each gene, color intensity represents the

number of flux samples per bin. Bottom: representative prediction score distributions for correctly and incorrectly predicted genes. **d** Top: reaction feature importance across all genes employed for training, using the random forest feature importance scores. Bottom: Importance of top 20 features across 50 repeats; box plots show mean, IQR, and whiskers are all samples not determined to be outliers. **e** Performance of FCL with smaller and less dense training data; error bars are mean ± standard error across $N = 5$ training repeats with different initializations. **f** Performance of FCL with earlier versions of the *E. coli* genome-scale metabolic model. Test results were computed across $N = 848$ genes shared by the four models[11]. Significance was determined at $p < 0.05$ with a one-sided *t*-test; error bars are the mean ± standard error across $N = 5$ training repeats. Only iJR904 was found to have statistically lower performance than iML1515 ($p = 0.0065$). Measured (ground truth) essentiality labels were taken from the literature[13]. Source data are provided as a Source data file.

mitigating the impact of poor GEM quality on the FCL predictive performance. When tested across different carbon sources, FBA delivers a maximal accuracy of 93.5% correctly predicted genes for *E. coli* growing aerobically in glucose with biomass synthesis as optimization objective[13]. We employed FCL using $N = 1202$ gene deletions (80%) with $q = 100$ samples/cone for training a binary classifier of gene essentiality; the biomass reaction was removed from training to prevent the model from learning the correlation between biomass and essentiality that support FBA predictions (Supplementary Fig. S3 and Supplementary Table S3). This led to a training dataset with $N = 120,285$ samples and $n = 2712$ features. We opted for a random forest classifier as a suitable compromise between model complexity and interpretability. Test results in a random set of $N = 300$ held-out genes (20%) outperformed the state-of-the-art FBA predictions in accuracy, precision and recall, achieving an average 95% accuracy for all test genes across training repeats (Fig. 2a and Supplementary Fig. S5); moreover, FCL achieved a 1% and 6% improvement in classification of nonessential and essential genes, respectively, as compared to FBA (Fig. 2b).

Inspection of sample-wise prediction scores show that a small number of deletions get incorrectly classified, likely due to GEM misspecifications (Fig. 2c). Interpretability analysis revealed that a few as 100 reactions can explain model predictions, with top predictors being enriched for transport and exchange reactions (Fig. 2d). Thanks to its excellent predictive power, FCL can be employed to define a distance metric between deletions and the wild type strain, with statistically significant differences between nonessential and essential deletions (Supplementary Fig. S4).

To investigate which factors determine FCL performance, we first retrained the model with sparser sampling data and fewer gene deletions (Fig. 2e); predictive accuracy dropped in both cases, but models trained on as few as 10 samples/cone already matched the current state-of-the-art FBA accuracy. We additionally retrained FCL with earlier and less complete GEMs for *E. coli* and found that only the smallest GEM (iJR904) displayed a statistically significant drop in performance (Fig. 2f). Given the high dimensionality of the feature space, we retrained the random forest model on a reduced feature set computed with Principal Component Analysis, but this resulted in lower accuracy

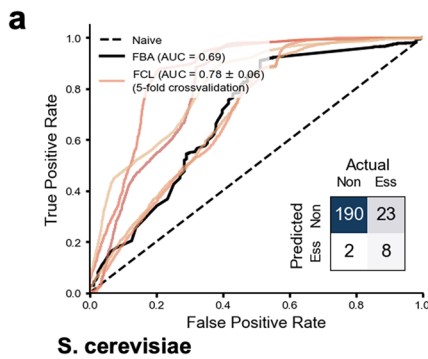

**S. cerevisiae**

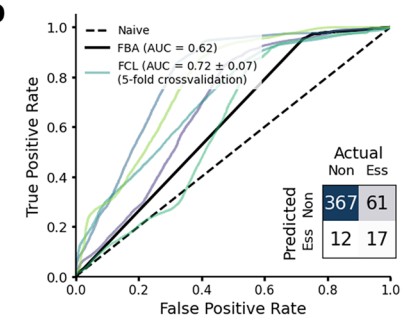

**Chinese Hamster Ovary**

**Fig. 3 | Prediction of metabolic gene essentiality for higher-order organisms.**
**a** Receiver operating characteristic (ROC) curves of FCL model for *Saccharomyces cerevisiae* using a random forest classifier and 5-fold cross-validation; models were trained on $N = 897$ class-stratified gene deletions with $q = 124$ samples/cone computed from the Yeast9 genome-scale model[33]. Solid black lines are FBA baseline predictions computed for all genes in the GEM. **b** ROC curves of FCL model for Chinese Hamster Ovary cells using a gradient boosting classifier and 5-fold cross-validation; models were trained on $N = 1832$ class-stratified gene deletions with $q = 127$ samples/cone computed from a well-adopted genome-scale model[32]. In

both panels, insets show confusion matrices for FCL predictions computed on class-stratified test genes held out from training (20%) and averaged across all folds. AUROC metrics for FCL models were computed as an average across folds ± one standard deviation. Solid black lines are FBA baseline predictions computed for all genes in the GEM. Details on model training can be found in the "Methods"; additional classification metrics can be found in Supplementary Table S5. Measured (ground truth) essentiality labels were taken from the literature[33,34]. Source data are provided as a Source data file.

in all tested cases, possibly because correlations between essentiality and small changes in cone shape can only be captured in a high-dimensional feature space. We also explored the use of deep learning models, including feedforward and convolutional neural networks, but these did not improve performance even when trained on larger data with more than $q = 5000$ samples/cone (not shown). This is likely because such models are deliberately overparameterized to accommodate highly nonlinear correlations among features, but in our case, flux samples are linearly correlated through the stoichiometric constraint in Eq. (1).

We tested FCL for essentiality prediction in *Saccharomyces cerevisiae* and Chinese Hamster Ovary (CHO) cells, two more complex organisms with well-curated GEMs[32,33] widely employed for the synthesis of heterologous proteins and metabolites. These models have 52% and 130% more reactions than *E. coli*, respectively, leading to a higher dimensionality of the flux cone and more features for training. Following the same data generation protocol as in Fig. 2a, we trained FCL on 80% of deleted genes with similar sample density and using essentiality labels from the literature[33,34]. FCL achieved better classification results than FBA in both organisms across multiple performance scores (Fig. 3 and Supplementary Table S5). We found that FCL showed similar prediction errors as FBA, with a tendency to misclassify some essential genes as nonessential, likely due to the class imbalance in the training data (most genes are nonessential). In the case of CHO cells, the performance gains over FBA are narrow, likely due to incomplete curation of the GEM; FCL performance could likely be improved further by designing GEM-specific machine learning architectures, particularly for large models such as those of CHO cells.

The performance improvements against FBA predictions on three organisms of varied complexity (*E. coli*, *S. cerevisiae*, and CHO cells) suggest that FCL provides the most accurate predictions for metabolic gene essentiality in the literature. This result further demonstrates that optimality assumptions are not required for prediction of metabolic gene essentiality, in agreement with earlier evidence provided by recent studies[19,35].

### Prediction of small molecule synthesis

To explore the power of FCL for predicting other phenotypes, we focused on small molecule biosynthesis in microbial strains engineered with heterologous pathways[36]. Recent studies have showcased the utility of genome-wide deletion screens for improving production

titers[37,38]. Nonessential deletions can both suppress or boost metabolite production; for example, deletions that disrupt enzymatic cofactor homeostasis are deleterious for product synthesis, while other nonessential deletions can redirect metabolic flux away from nonessential pathways toward increased production[3].

We focused on a large deletion screen of *S. cerevisiae* mutants engineered to synthesize betaxanthin[37], a tyrosine-derived pigment widely employed in the food sector. The screen includes a total of 4223 gene deletions, out of which $N = 811$ genes code for metabolic enzymes present in the latest yeast GEM[33]. Fitness scores for each deletion strain were quantified via betaxanthin autofluorescence averaged across four nonclonal cultures (Fig. 4a). We first binned betaxanthin autofluorescence into three classes for low, medium, and high-producing cultures. We employed FCL to build a 3-class classifier that predicts betaxanthin synthesis using Monte Carlo sampling of the deletion GEMs. Due to the imbalanced data size across classes (17.1%, 67.2%, and 15.7%, respectively), we trialed various model architectures in combination with rebalancing strategies (Fig. 4b); the best-performing model delivered promising accuracy (69.8%). We observed a tendency to underpredict the high-producing deletions due to these being underrepresented in the training data, though high producer accuracy improvements between 5.5% and 28.3% could be obtained via various class balancing techniques (Fig. 4c and Supplementary Table S6).

To the best of our knowledge, this is the first demonstration that small molecule synthesis can be predicted from deletion screening data, and adds to the growing number of tools to predict metabolite production using various data modalities and computational approaches[39–41]. Since FCL relies purely on the wild-type GEM and experimental fitness readouts, it does not require extending the GEM with a heterologous pathway, which is particularly beneficial for production pathways with poorly characterized stoichiometry.

## Discussion

With the rapid progress in high-throughput genetic engineering and automated screening technologies, there is a growing opportunity to utilize such data for building predictors of the phenotypic response to gene deletions. FCL offers a general strategy to detect correlations between metabolic genotypes and phenotypic readouts. It combines experimental fitness data with mechanistic knowledge into a machine learning system able to draw phenotypic predictions for a specific gene deletion.

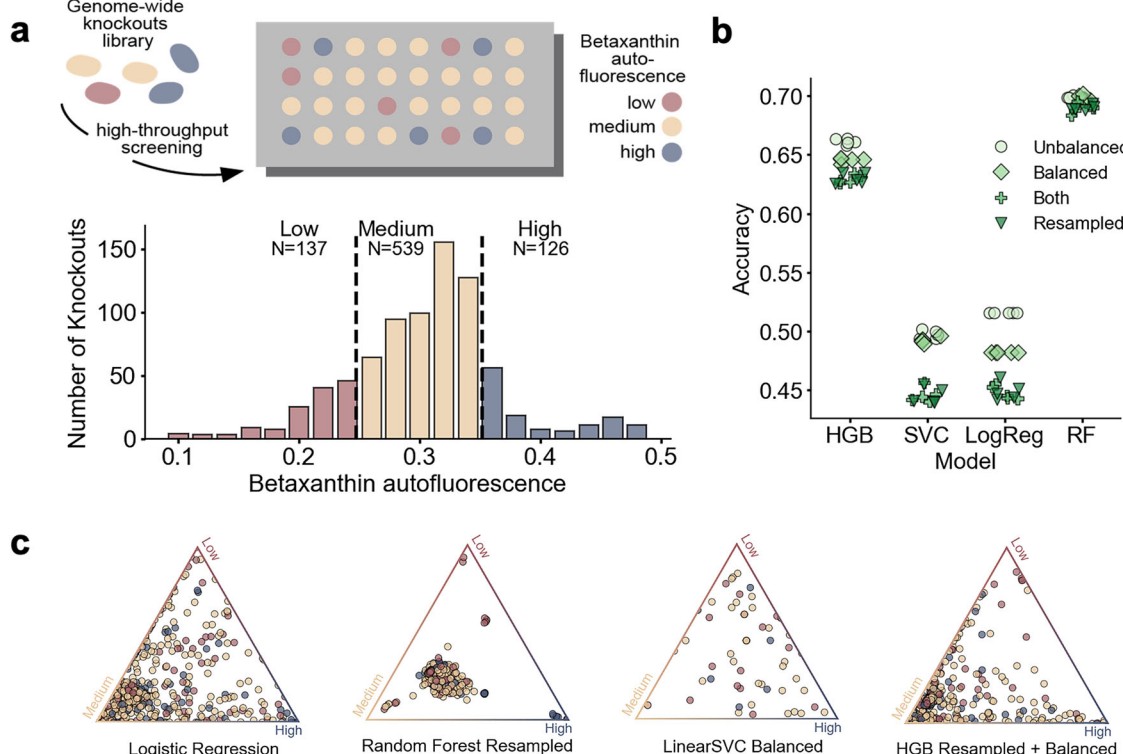

**Fig. 4 | Prediction of small molecule synthesis in *Saccharomyces cerevisiae*.**
**a** High-throughput deletion screening data of *S. cerevisiae* strains engineered to produce betaxanthin[37]; fitness scores were determined from betaxanthin autofluorescence, normalized and binned into three classes for model training. The majority of deletions are medium producers, with ~15% of deletions being high producers. **b** Accuracy results for several FCL models with different algorithms for multiclass classification of deletion strains (HGB Histogram-based Gradient Boosting, SVC Support Vector Classifier, LogReg Logistic Regression, RF random forest) and various strategies to rebalance the three classes. Accuracy was computed across all three classes and shown for $N = 5$ training repeats. Balanced: class labels were weighted to account for class imbalance; Resampled: majority class was subsampled to be the same size as the minority classes. Class rebalancing can increase accuracy for high producers, often at the expense of overall accuracy. For full class balancing results on a held-out 20% test set ($N = 659$ deletions), see Supplementary Table S6. **c** Ternary plots of model predictions on the test set for representative models with varying predictive accuracy across the three classes. Vertices represent class prediction with probability one (full confidence), whereas central points are deletions predicted to be equally likely to be any of the three classes. Each sample has been color-coded according to its ground truth class labels. Source data are provided as a Source data file.

Our model evaluations demonstrate that FCL outperforms the state-of-the-art FBA predictions of metabolic gene essentiality. FBA has the advantage of being a zero-shot predictor, in the sense that it does not need to be trained on fitness data. Instead, FBA draws predictions based on a biological optimality assumption; for microbial systems, maximal growth rate or biomass synthesis rate are well-validated metabolic objectives. But for most organisms beyond the microbial world, such optimality assumptions are not warranted, and there is no consensus on how to define suitable metabolic objectives for higher-order organisms[16,18]. Various studies have built strategies to accommodate the multiobjective nature of metabolic optimality[42,43] or to reverse engineer metabolic objectives[44,45] and tradeoffs[18,46]. Yet even in cases where an optimal objective of the wild-type can be validated, there is little evidence that such an objective would be preserved upon a gene deletion. Mutants are likely to be subject to different evolutionary pressures that shift their genetic programs away from the physiological objectives of the wild-type. FCL thus allows essentiality predictions in a much wider range of cell types than current methods, including those with unknown optimality principles such as human cell lines[47] or the gut microbiome[48], as well as prediction of other deletion phenotypes beyond essentiality, such as single-cell metabolic capabilities[49], synthetic lethality[50], or gain-of-function deletions[51]. Although FCL is agnostic to the fitness score employed for training, its effectiveness is limited by the strength of correlations between metabolic activity and the phenotype of interest. In the case of gene essentiality, for example, FCL works well because deletions in pathways that supply key metabolites for growth can strongly impact cell viability. Other phenotypes with weaker or no associations to metabolic activity may require additional data modalities for accurate prediction.

The integration of learning algorithms with GEMs has shown substantial promise for improved predictivity across various tasks[18,19,41,52–54]. The novel paradigm behind FCL is to learn the shape of the metabolic space through random sampling of GEM. High-dimensional sampling remains a key challenge in statistical learning[55], because in high dimensions, samples tend to be equidistant and concentrate on the boundaries of the space[56]. While expectation would suggest that dense sampling is needed to accurately capture the cone geometry, we consistently found that accurate FCL models could be trained from shallow sampling with as few as 100 samples per deletion. We hypothesize this is a case of the curse-of-dimensionality working to our benefit: to capture changes to the cone, FCL only requires samples at the boundary, and therefore, a relatively small number of samples is sufficient for accurate prediction.

An exciting application of FCL is the discovery of knockouts or deletion strategies that result in improved production of small molecules. This would help reduce the number of costly experiments required for strain optimization. A key challenge, however, is that desirable traits such as high metabolite titer are rare, which results in substantial class imbalances like those observed in the betaxanthin dataset (Supplementary Table S4); only a few knockouts improve production, and therefore, training data is typically enriched for mid- or low-producers. Data augmentation and synthetic data generation could address some of these challenges, in addition to new model

architectures that improve performance. We also note that when designing FCL-based machine learning models, the experimental reproducibility of production readouts should put a ceiling on the expected model accuracy, so as to avoid training models that predict with higher accuracy than the measurement error.

The performance of FCL suggests that predictive representations of metabolic capabilities can be learned from Monte Carlo sampling of GEMs. This advancement lays the groundwork for the development of metabolic foundation models via large sampling across species, growth conditions and deletion genotypes, thus extending the breadth of biological foundation models across additional layers of cellular organization[57,58]. We expect that FCL will open new routes for computational prediction of many cellular phenotypes, with applications in basic discovery, biotechnology and future therapies.

## Methods

FCL utilizes sampling data generated with a random walk on a deletion-specific GEM. First, a wild-type GEM is modified with a gene deletion by setting the corresponding reaction bounds to zero. The high-dimensional flux cone of the deletion GEM is sampled using a random walk sampler; in our implementations, we opted for OptGPSampler[59], a fast Monte Carlo method that aims to uniformly sample the flux cone. The sampler first transforms the problem into a convex optimization problem in logarithmic space using geometric programming, then employs a hit-and-run algorithm to sample the interior of the cone. The resulting flux data have a number of rows equal to the number of samples and a number of columns equal to the number of reactions in the GEM. Each deletion produces a collection of flux sampling vectors, all of which are labeled with a fitness score obtained from experimental data. The fitness score can be either discrete or continuous, depending on the fitness readout under study.

The resulting labeled dataset is then employed for training supervised machine learning models. The model predictions are made at the level of flux samples, i.e., one row of the flux sampling data frame from each deletion is passed through the trained machine learning model to produce a predicted fitness score. Therefore, every sample from each deletion GEM is assigned an individual predicted score, and the distribution of these scores is finally averaged to obtain a gene-level prediction. FCL can deliver high predictive accuracy because it is trained to learn correlations between the geometry of the flux cone and the resulting phenotype.

### Generation of flux sampling data

Flux sampling is a collection of methods for randomly generating flux distributions from the solution space of a GEM. Flux sampling algorithms are based on random walks optimized for the high-dimensional and nonisotropic geometry of the convex polytope defined by the GEM.

OptGPSampler uses artificial centering hit-and-run to bias the random walk towards the elongated regions of the flux cone. After an initial random location in the flux space is selected and a warm-up phase, every $k$th point following is generated by the sampler until $N$ points are generated. These two parameters ($k, N$) control the number of flux samples generated by the algorithm. Flux sampling is computationally costly because it requires running a random walk on a high-dimensional flux space that needs to reach mixing time to achieve uniform coverage.

We ran OptGPSampler on all single-gene deletions in four *E. coli* models, the Yeast9 model for *S. cerevisiae*, and the iCHO2291 model for CHO cells. For training supervised machine learning models, sampling data were normalized to zero mean and unit variance. There were a small number of deletions in each GEM where the sampling failed to converge; these were not included in training or testing. A summary of GEM sizes and sampling data can be found in Supplementary Tables S1, S2. For example, in the Yeast9 model, we sampled 1159 single-gene

deletions with a step size of $k = 5000$ for a sampling density of $N = 124$ samples/cone, leading to a total of 143,716 samples with $D = 4130$ fluxes each (total data size 4.43 Gb).

For all models except *E. coli*, we sampled with a high step size of $k = 5000$. To ensure robust performance evaluations in the *E. coli* iML1515 model (Fig. 2a–d), we retrained models many times using different training sets. For computational efficiency and due to large data sizes, after computing an initial large set of samples, using a fine step size of $k = 100$, we subsampled the data 10 times to have the same number of samples per deletion ($N = 100$ samples/cone). Three smaller *E. coli* models (iAF1260, iJO1366, iJR904) were employed for the comparison in Fig. 2f. In these models, deletion GEMs were sampled with $N = 100$ samples/cone and $k = 5000$. To equalize the amount of training features between models, only the deletions present in all models ($D = 864$ reactions) were included in the training and test sets for the models in Fig. 2f. The biomass reactions were removed to ensure the models were learning from the true reaction fluxes, not the biomass reaction used to compute FBA predictions (see Supplementary Table S3).

### Experimental fitness labels

The gene essentiality labels for *E. coli*, *S. cerevisiae* and CHO cells were obtained from the literature[13,33,34]. The yeast labels included nonmetabolic genes and were labeled with both gene and ORF labels. Gene names were standardized to their systematic names from the Saccharomyces Genome Database, resulting in $N = 1121$ metabolic gene deletions labeled with essentiality data, sampled, and included in the final dataset for model training. ORFs and gene names were linked using a tool from Yeastract+ http://www.yeastract.com/formorftogene.php.

For the results in Fig. 4, betaxanthin autofluorescence readouts for $N = 811$ yeast deletions were taken from Cachera et al.[37] and averaged across four cultures. While one gene (YBR011C) was also identified as essential in other studies, we included it in our analysis as we hypothesized this could be a conditionally essential deletion which can grow in alternative strain and media conditions. The average autofluorescence was normalized to the (0,1) range. We first framed the problem as a regression task, but this proved challenging with the limited number of knockouts at the high and low ends of the autofluorescence distribution. Recognizing that predicting high or low producers is a core task in several applications, we chose to train a three-class classifier by binning the data into three classes of high, medium, and low producers (Fig. 4a). We set the thresholds qualitatively to label 67% of samples as medium producers (within -1 standard deviation from the mean). In all our case studies, labels were highly class-imbalanced, as shown in Supplementary Table S4.

### Training of supervised learning models

All models were trained using the scikit-learn package in Python.

*Escherichia coli*. A random forest model classifier was trained on an 80% training set stratified to maintain the class imbalance. Model hyperparameters were fixed as `max_depth` =None, `min_samples_split` =2. The random forest was retrained $N = 5$ times with different test sets to confirm that performance was not significantly affected by the composition of the training set. The FBA baseline was obtained using the `single_gene_deletion` function in the CobraPy package[60] applied to all genes in the iML1515 model with default biomass objective function, aerobic conditions, and glucose as the carbon source. For the results in Fig. 2a, b, we chose 0.41/h as the cutoff for FBA predictions. This was chosen to match the experimental growth rate cutoff employed by the original iML1515 source[13], which is 50% of the wild-type growth rate (predicted to be -0.81/h by FBA). A ROC curve computed across five repeats is included as Supplementary Fig. S5, which demonstrates that FCL outperforms FBA in *E. coli*

essentiality prediction regardless of the chosen cutoff. The naive baseline was compared by predicting all genes as nonessential (majority class). Once trained on the sample level, the prediction score of all samples from a single deletion was averaged; if this score was less than 0.5, the deletion was classified as essential. A representative model was used to create the prediction score distributions in Fig. 2c. In Fig. 2d, we trained $N = 50$ random forest classifiers on one sub-sample with random held-out test sets. Feature importance scores of all reactions were extracted from the random forest models.

*Saccharomyces cerevisiae*. For essentiality prediction, a class-stratified 20% of deletions (192 nonessential; 31 essential) was held out as a test set. The remaining 80% of deletions (772 nonessential; 126 essential) were split into 5-fold cross-validation sets and a random forest model was trained on each fold. The `max_depth`, `n_estimators`, and `min_samples_split` hyperparameters were tuned using a grid search and the model with the highest average cross-validation accuracy was selected, and the confusion matrix and ROC curve were computed for Fig. 3a. The best `max_depth` value was 30, the best `n_estimators` value (the number of trees) was 300, and the best `min_samples_split` (the minimum number of data points to split a leaf on the random forest) value was 2. The minimum deletion-level accuracy was 87.5%, the maximum was 90.3%. The test set results were computed by running the held-out test set through all 5-fold models and averaging the deletion-level scores across all models. The FBA baseline was computed using the `single_gene_deletion` function in Cobrapy for all genes with glucose as the carbon source and the standard biomass reaction.

For prediction of betaxanthin synthesis (Fig. 4), multiple models were trained on a class-stratified 80% training set split with a consistent held-out test set. The following model types were trained: Hist-GradientBoostingClassifier, Linear Support Vector Classifier, Logistic Regression Classifier, and Random Forest Classifier. We implemented two class balancing techniques to improve the minority class performance: balancing, which weights the class labels to account for the class imbalance, and resampling, which subsamples the majority class to be the same size as the minority classes.

**Chinese hamster ovary cells.** A HistGradientBoosting Classifier was trained on a 5-fold cross-validation of the training set. Twenty percent of the original training dataset was held out as a test set and not included in the cross-validation. The large training set data size required training models across 4 CPU nodes to load all training data into memory. The hyperparameters `learning_rate`, `max_iter`, and `max_depth` parameters were tuned via grid search and the model with the highest average cross validation accuracy was selected and the confusion matrix and ROC curve computed for Fig. 3b. The `learning_rate` was varied between 0.01 and 0.2, the `max_iter` between 100 and 500, and the `max_depth` set to 5, 10, or None. The best model had a `learning_rate` of 0.05, a `max_iter` of 100, and a `max_depth` of None. The test set results were computed by running the held-out test set through all 5-fold models and averaging the deletion-level scores across all models. The confusion matrix was computed for a class threshold value of 0.5 for each fold, and counts were averaged across all 5 folds. The FBA results were computed using the *single_gene_deletion* function in Cobrapy for all deletions and the default carbon source and objective function in the iCHO2291 model.

### Reporting summary
Further information on research design is available in the Nature Portfolio Reporting Summary linked to this article.

### Data availability
Flux sampling data and experimental fitness labels employed in the paper have been deposited in Zenodo[61] at https://doi.org/10.5281/ zenodo.15518666. Due to data size, we have provided one set of flux samples for each of GEMs employed in the paper. These can be employed to retrain models presented in the paper with the code provided. Source data are provided with this paper.

### Code availability
The code used to train the models, perform the analyses and generate results in this study is publicly available and has been deposited in Zenodo[61] under license CC-BY 4.0. The specific version of the code associated with this publication can be found at https://doi.org/10. 5281/zenodo.15518666.

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

## Acknowledgements

C.M. and D.A.O. were supported by the United Kingdom Research and Innovation (grant EP/S02431X/1, UKRI Centre for Doctoral Training in Biomedical AI).

## Author contributions

C.M. performed data analysis, model training, and benchmarking. O.M.A. advised on machine learning aspects. DAO designed the research and provided overall supervision.

## Competing interests

The authors declare no competing interests.
