## [Transparent Peer Review file · Nature Communications]

Accurate prediction of gene deletion phenotypes with Flux Cone Learning

Corresponding Author: Professor Diego Oyarzún

Version 0:

Reviewer comments:

Reviewer #1

(Remarks to the Author)

In this paper Merzbacher, Aodha, and Oyarzun develop a novel framework for predictive modelling of metabolic engineering methods (e.g. gene deletion) applied to cell growth, and assess this on several model organisms (*E. coli*, *S. cerevisiae*, CHO). Their paper presents and develops a novel idea, tackling an outstanding challenge in a widely studied field which will be of broad general interest. This challenge is one that has been thought about and approach from many angles in the past, and the authors build upon this progress while providing compelling evidence that the method they developed may outperform existing state-of-the-art on several key benchmarks. The methodology and results are well explained, with communication clear and effective throughout the manuscript. Overall I am therefore highly supportive of its publication, with a few minor comments below that the authors could consider in their review process.

Line 58-64. This explanation of the algorithm is core to the paper, yet on first reading I found it somewhat difficult to grasp the different steps. For example, perhaps explanation could be expanded slightly e.g. explaining in broad picture terms what it means to do monte-carlo sampling of a flux cone, what the resultant data "is" and how it can then be used to train a model. I recognise this is illustrated to some extent in Figure 1 but, at least in my opinion, some explanation-via-example might benefit the narrative here for readers that aren't deep in this field already.

Line 146. Here it is stated this is the first predictive tool for deletion screening data to predict small molecule synthesis. Perhaps this is true technically, but are there not other similar works which might be included here/elsewhere in the manuscript as background? For example: <https://jcheminf.biomedcentral.com/articles/10.1186/s13321-018-0324-5> Admittedly these are significantly different approaches, but nevertheless may be relevant background for the work.

Line 149. The statement that "[FCL relies on wild-type GEM without extension with heterologous pathway]... suggests that FCL can be an effective tool for data-driven strain design". Is there more nuance to this statement? Surely there are important cases where it would be better/only possible by considering properties of the heterologous pathway?

Line 332. Here (and elsewhere) fitness scores from experimental data are referred to and suggesting that the method is somewhat agnostic to these. Is that the case? Is it worth adding a couple of sentences here relating to whether some fitness scores/experimental data types may be more or less viable? Further, there is a typo on this line "with a fitness score obtained FROM experimental data." I think I spotted a few typos elsewhere while reading so it would pay to do a careful edit.

Line 387. Here the method for binning autofluorescence is explained - would it be possible to motivate this? For example why three classes, and why are cut-offs set as they are? (e.g. it isn't done to balance the N in each class as far as I can see, is the choice of 0.25 and 0.35 essentially arbitrary?) I am not expecting the authors to re-do these simulations with different classification approach, but it may be worth at least discussing in the methods.

Congratulations again to the authors on a really nice paper!

(Remarks on code availability)

Reviewer #2

(Remarks to the Author)

The manuscript presents Flux Cone Learning (FCL), a novel framework aimed at predicting the phenotypic effects of metabolic gene deletions. FCL combines high-dimensional Monte Carlo sampling of the metabolic flux space with supervised machine learning, claiming superiority over traditional Flux Balance Analysis (FBA) in predicting gene essentiality across organisms such as *Escherichia coli*, *Saccharomyces cerevisiae*, and Chinese Hamster Ovary (CHO) cells. Additionally, the method extends to predicting small molecule production from gene deletion screens.

Major Comments

1. Novelty and Comparison to Existing Methods

(1) The FCL framework (flux-sampling + ML) is not fundamentally new: FluxGAT (arXiv:2403.18666) already employs flux sampling together with graph neural networks for the same goal. The only substantive difference is the choice of learner (classical supervised models vs. GNNs).

(2) On the CHO cell essentiality benchmark (reference 47), FCL achieves AUC = 0.72 versus FluxGAT's 0.765, and even underperforms its own baselines (node2vecGAT, GCN). The manuscript lacks any direct experimental head-to-head with FluxGAT, gMCS, and other established gene-essentiality predictors. Without these comparisons, it is impossible to judge whether FCL offers a genuine advance.

(3) The introduction lacks a comprehensive review of existing gene essentiality prediction tools, including FluxGAT, gMCS, and non-metabolic network-based methods, which weakens the claim of novelty.

2. Evaluation Metrics and Statistical Rigor

(1) Results are reported only in terms of accuracy and ROC-AUC. Standard ML practice requires at minimum precision, recall, and F1-score—especially where class imbalance may skew accuracy.

(2) No confidence intervals or statistical tests are provided to assess whether reported gains over FBA (or other methods) are significant.

3. Objective Function in Flux Sampling

(1) The authors claim FCL “does not require an objective function,” yet their sampling procedure necessarily uses the model's default objective (typically biomass maximization). They do not explore how changing the objective—for instance to β -carotene production—would alter sampling outcomes or prediction performance.

(2) For the β -carotene classification task, it is unclear whether the flux samples were drawn under a biomass objective or a β -carotene objective. A direct comparison—sampling under each objective, and even FBA-based essentiality prediction for β -carotene—would clarify the method's sensitivity to this choice.

4. Thresholds and Parameter Justification

Key experimental details are missing, such as the objective function used during flux sampling for β -carotene synthesis predictions and the rationale for the growth rate threshold of 0.4 h^{-1} for gene essentiality classification. This threshold deviates from the standard 5-10% of wild-type growth rate typically used in GEM-based predictions, and its justification is unclear.

5. Code Availability

The provided code link fails (“MultCloud is not supported in this region”), preventing validation. Please host code in a publicly accessible repository (e.g. Zenodo) and include clear installation and usage instructions.

(Remarks on code availability)

The provided code link fails (“According to local laws, MultCloud is not supported in this region.”), preventing validation.

Version 1:

Reviewer comments:

Reviewer #1

(Remarks to the Author)

Thank you for addressing all of my suggested comments, and congratulations on an improved manuscript.

(Remarks on code availability)

Reviewer #2

(Remarks to the Author)

The authors have responded to most of the concerns raised during the first round of review in a careful and constructive manner, for which I am appreciative. However, after reviewing the revised manuscript and the authors' responses in detail, I find that several key issues remain unresolved and warrant further clarification, justification, or discussion:

1. On the Threshold for Defining Gene Essentiality (Residual Issue from Round 1):

The authors continue to define gene essentiality using a cutoff of 0.4 h^{-1} for growth rate reduction, citing Monk et al. (2017) as the source. However, the supplemental materials of the iML1515 paper (Monk et al., 2017, lines 523–524) clearly state that: "A gene was considered computationally essential for the simulated condition if deletion of the gene reduced optimal growth rate to less than 0.05 h^{-1} ."

This raises a significant concern: the 0.4 h^{-1} threshold used by the authors deviates substantially from the original 0.05 h^{-1} . This discrepancy has serious implications for the classification of essential genes, and thereby the reliability and comparability of downstream analyses.

2. On the Significance of Cross-Species Performance Improvements:

The authors evaluated the performance of their method in predicting essential genes across three species: *Escherichia coli*, *Saccharomyces cerevisiae*, and Chinese Hamster Ovary (CHO) cells (Table S5). The results showed a significant improvement in *S. cerevisiae* (from 81% to 90%, a 9% increase), a modest improvement in *E. coli* (approximately 1.5%), and almost no improvement in CHO cells. Overall, the method yielded only limited performance gains across different species.

3. On the Practical Utility and Validation of Small Molecule Synthesis Prediction:

Figure 4B shows that SVG, LogReg, RF methods achieve accuracy close to or below 50%, essentially equivalent to random guessing. The HGB method HGB achieves ~63–67% accuracy (as visually estimated, though the manuscript reports 69.8% — please confirm consistency with the figure). In biological design or metabolic engineering, is this level of accuracy (~70%) sufficient to guide decision-making? A deeper discussion on the practical significance and limitations of this result is warranted. If possible, the authors are strongly encouraged to select a few representative predictions and carry out wet-lab validation experiments. Even a limited number of validations would significantly strengthen the manuscript's impact and demonstrate real-world applicability.

(Remarks on code availability)

The organization of the data and code is well-structured and clearly presented, which greatly facilitates understanding and reproducibility.

Version 2:

Reviewer comments:

Reviewer #2

(Remarks to the Author)

The authors have provided a relatively detailed response to the previously raised concerns, and their explanations on related issues are reasonably sound.

Regarding the threshold for defining gene essentiality, the authors clarified the rationale behind selecting a threshold of 0.4 h^{-1} . Through analyses such as ROC curves and FBA-predicted growth rate distributions, they demonstrated that the superiority of FCL on *E. coli* is not significantly affected by the choice of threshold. Moreover, the difference between the employed threshold and the recommended one appears to be minimal and attributable to data characteristics. This part of the response adequately addresses concerns about threshold selection.

In terms of cross-species performance, the authors explained the performance differences across species. For instance, the limited improvement in *E. coli* is due to its already high baseline accuracy, while the modest gains in CHO cells may relate to the completeness of their genome-scale models. These explanations are reasonably convincing.

However, from a practical application standpoint, the predictive performance of FCL remains somewhat limited. Compared to FBA, the improvements in certain species (e.g., *E. coli*, CHO cells) are not particularly significant. As the authors noted, data bias may be one of the main contributing factors. In addition, although the authors discussed the influencing factors on prediction accuracy for small molecule biosynthesis, the lack of wet-lab validation weakens the practical value and impact of the method, falling short of being compelling or groundbreaking.

(Remarks on code availability)

Response to referees

Accurate prediction of gene deletion phenotypes with Flux Cone Learning

Charlotte Merzbacher, Oisín Mac Aodha, Diego A. Oyarzún

Reviewer #1 (Remarks to the Author):

Q: In this paper Merzbacher, Aodha, and Oyarzun develop a novel framework for predictive modelling of metabolic engineering methods (e.g. gene deletion) applied to cell growth, and assess this on several model organisms (E. coli, S cerevisiae, CHO). Their paper presents and develops a novel idea, tackling an outstanding challenge in a widely studied field which will be of broad general interest. This challenge is one that has been thought about and approached from many angles in the past, and the authors build upon this progress while providing compelling evidence that the method they developed may out-perform existing state-of-the-art on several key benchmarks. The methodology and results are well explained, with communication clear and effective throughout the manuscript. Overall I am therefore highly supportive of its publication, with a few minor comments below that the authors could consider in their review process.

A: We thank the reviewer for the encouraging assessment of our work.

Q: Line 58-64. This explanation of the algorithm is core to the paper, yet on first reading I found it somewhat difficult to grasp the different steps. For example, perhaps explanation could be expanded slightly e.g. explaining in broad picture terms what it means to do Monte Carlo sampling of a flux cone, what the resultant data "is" and how it can then be used to train a model. I recognise this is illustrated to some extent in Figure 1 but, at least in my opinion, some explanation-via-example might benefit the narrative here for readers that aren't deep in this field already.

A: We thank the referee for this comment. We agree that further elaboration is needed and have added additional explanation and an example in the text [L83-88]. The feature matrix for model training has $k \times q$ rows and n columns, where k is the number of gene deletions, q is the number of flux samples per deletion, and n is the number of reactions in the GEM. We have modified Figure 1 to better reflect the format of the flux sampling data.

Q: Line 146. Here it is stated this is the first predictive tool for deletion screening data to predict small molecule synthesis. Perhaps this is true technically, but are there not other similar works which might be included here/elsewhere in the manuscript as background? For example: <https://jcheminf.biomedcentral.com/articles/10.1186/s13321-018-0324-5>

Admittedly these are significantly different approaches, but nevertheless may be relevant background for the work.

A: We agree and apologize for this omission. We have included the requested reference plus other relevant pointers to literature (L165-168).

Q: Line 149. The statement that "[FCL relies on wild-type GEM without extension with heterologous pathway]... suggests that FCL can be an effective tool for data-driven strain design". Is there more nuance to this statement? Surely there are important cases where it would be better/only possible by considering properties of the heterologous pathway?

A: We agree that our statement was too broad and have rephrased it in L168-170 to highlight the benefits of FCL for cases where pathway stoichiometry is not well characterized. Certainly there are strain engineering problems that can only be solved by examining the heterologous pathway in detail.

Q: Line 332. Here (and elsewhere) fitness scores from experimental data are referred to and suggest that the method is somewhat agnostic to these. Is that the case? Is it worth adding a couple of sentences here relating to whether some fitness scores/experimental data types may be more or less viable? Further, there is a typo on this line "with a fitness score obtained FROM experimental data." I think I spotted a few typos elsewhere while reading so it would pay to do a careful edit.

A: Thank you for pointing out this nuance. We agree and have rephrased L195-200 accordingly. Although FCL is indeed agnostic to the fitness score employed for training, its effectiveness is limited by the strength of correlations between metabolic activity and the phenotype of interest. In the case of gene essentiality, for example, FCL works well because deletions in pathways that supply key metabolites for growth can strongly impact cell viability. Other phenotypes with weaker or no associations to metabolic activity may require additional modalities of data for accurate prediction. We have also corrected the typo.

Q: Line 387. Here the method for binning autofluorescence is explained - would it be possible to motivate this? For example why three classes, and why are cut-offs set as they are? (e.g. it isn't done to balance the N in each class as far as I can see, is the choice of 0.25 and 0.35 essentially arbitrary?) I am not expecting the authors to re-do these simulations with different classification approach, but it may be worth at least discussing in the methods.

A: We agree with the reviewer that further explanation is needed and have added additional sentences to the methods (L433-438). The average autofluorescence was first normalized to the range (0, 1) from a raw range of (0.28, 0.61). Initially we framed the problem as a regression task, but this proved challenging with the limited number of samples at the high

and low ends of the autofluorescence distribution. Recognizing that predicting extremely high or low producers is a core task for experimentalists, we chose to bin the data into three classes of high, medium, and low producers. We set the thresholds qualitatively to label approximately the ± 1 SD region (67% of samples) as medium producers and samples above and below this middle region as high or low.

Q: Congratulations again to the authors on a really nice paper!

A: Thank you for the enthusiastic comments.

Reviewer #2 (Remarks to the Author):

Q: The manuscript presents Flux Cone Learning (FCL), a novel framework aimed at predicting the phenotypic effects of metabolic gene deletions. FCL combines high-dimensional Monte Carlo sampling of the metabolic flux space with supervised machine learning, claiming superiority over traditional Flux Balance Analysis (FBA) in predicting gene essentiality across organisms such as *Escherichia coli*, *Saccharomyces cerevisiae*, and Chinese Hamster Ovary (CHO) cells. Additionally, the method extends to predicting small molecule production from gene deletion screens.

A: Thank you for the accurate summary of our manuscript.

Q: Novelty and Comparison to Existing Methods

(1) The FCL framework (flux-sampling + ML) is not fundamentally new: FluxGAT (arXiv:2403.18666) already employs flux sampling together with graph neural networks for the same goal. The only substantive difference is the choice of learner (classical supervised models vs. GNNs).

A: Thank you for this comment. FluxGAT is an elegant extension of our previous FlowGAT model (Hasibi et al, NPJ Syst Biol Appl, 2024), which in turn is based on the Mass Flow Graph construction we first proposed (Beguerisse et al, NPJ Syst Biol Appl, 2018). We have now included a reference to FluxGAT and apologise for this omission.

Flux Cone Learning (FCL) differs from FluxGAT in substantial ways and not just in the choice of learner. First, we clarify that FluxGAT is not trained on flux sampling data, but on a single flux vector representing the centroid of the flux cone and computed as the average vector across flux samples. FCL instead is trained on the thousands of sampled flux vectors themselves, which allows it to capture correlations between subtle changes to cone geometry and the phenotype of interest.

Second, by construction FluxGAT predicts *reaction* essentiality and not *gene* essentiality. This has important implications for training and the applicability of FluxGAT. Because

FluxGAT operates on the space of reaction fluxes, it cannot be trained directly on experimental gene deletion data. Gene labels first need to be mapped into reaction labels through the gene-protein-reaction (GPR) map. But as explained in Equation (8) of the FluxGAT paper, this conversion is challenging and not one-to-one: multiple genes can map to a single reaction and vice versa via AND and OR relationships. In simple cases where a single gene codes for a single reaction, gene labels map one-to-one onto reaction labels. Likewise, in cases where two genes affect a reaction via an OR relation and one is essential and the other nonessential, the reaction is considered nonessential. But in more complex relationships with a mix of ANDs and ORs, logical precedence must be followed to manually resolve the conflict. In other cases, some genes simply cannot be mapped onto reaction labels. This label conversion step limits the applicability of FluxGAT and introduces bias in the training data. For example, in the FluxGAT paper the label conversion step changes the class imbalance from 17% in the gene space to 25% in the reaction space.

Moreover, once FluxGAT is trained, the reverse problem must also be solved to convert reaction predictions into gene predictions. This backward conversion is lossy because the GPR map is not fully invertible and only few reactions can be mapped one-to-one back onto genes. For example, as mentioned in the FluxGAT paper, the algorithm can deliver predictions for only 752/2,085 genes present in the CHO model. A key implication of this is that FluxGAT predictions cannot be compared head-to-head with the state-of-the-art predictions of Flux Balance Analysis or other *gene* essentiality predictors.

Our own earlier algorithm FlowGAT (Hasibi et al, NPJ Syst Biol Appl, 2024), which precedes FluxGAT, also suffers from this deficiency; this was a key driver for us to step back from the graph-based approach and move toward the FCL sampling-based strategy. Flux Cone Learning completely resolves these issues because it is trained directly on gene-level fitness scores, without the need to convert labels through the GPR map. This allows using training data directly from knockout screening assays, without any pre- or post- processing of essentiality labels.

Q: (2) On the CHO cell essentiality benchmark (reference 47), FCL achieves AUC = 0.72 versus FluxGAT's 0.765, and even underperforms its own baselines (node2vecGAT, GCN). The manuscript lacks any direct experimental head-to-head with FluxGAT, gMCS, and other established gene-essentiality predictors. Without these comparisons, it is impossible to judge whether FCL offers a genuine advance.

A: In our manuscript we compare FCL head-to-head with FBA, the gold standard in the field and the most widely adopted method for predicting metabolic gene essentiality. We show that FCL outperforms FBA in *Escherichia coli*, *Saccharomyces cerevisiae* and *Chinese Hamster Ovary* cells, all organisms of varied complexity, with well curated GEMs and for which FBA predictions are generally considered of high quality. We believe this provides robust evidence that FCL offers a genuine advance and have modified the Abstract to highlight this contribution.

There is a confusion with regards to the classification scores for CHO predictions of FluxGAT. The mentioned AUC values are not comparable because they score different performance curves: the AUC reported for FluxGAT (0.765) was computed on the precision-recall curve (PRAUC), while our AUC score for FCL (0.72) was computed on the receiver operator characteristic curve (ROC-AUC).

To resolve this, we have now computed the precision-recall curves for FCL (Figure 1 below) and obtained an average AUC = 0.91 ± 0.03 in cross-validation, which outperforms both FluxGAT and FBA predictions. We remark that the comparisons against FluxGAT are qualitative in nature, because as mentioned earlier, FluxGAT evaluations were computed on reaction ground truth labels that have been filtered through the GPR map, while FCL and FBA evaluations are computed directly on gene labels. Moreover, performance may also be distorted by the reduction in the class imbalance introduced by the gene-to-reaction label mapping needed in FluxGAT.

Figure 1: Precision-recall curve of Flux Cone Learning (5-fold cross-validation) and FBA for prediction of gene essentiality in the CHO model.

We also trialed gene minimal cut sets (gMCS) as suggested by the reviewer. gMCS was specifically designed to predict synthetic lethal genes in cancer cell lines (Apaolaza et al., 2017, Valcárcel et al., 2024). A gMCS is the smallest set of genes that, when simultaneously removed, can disable a prescribed metabolic task such as growth. To predict essentiality, gMCS used RNA-seq data that has been thresholded into "high" and "low" expression categories to identify gMCS that contain highly expressed genes surrounded by lowly expressed genes. If one identifies a gMCS in which the target gene is the only highly expressed gene and the remaining genes are lowly expressed, then knockout of this target

gene would be sufficient to disable the metabolic function (Olaverri-Mendizabal et al., 2024).

We attempted to implement gMCS for our held-out test set of N=451 genes (Figure 3B), using the RNA-seq data that was previously used to parameterize the CHO genome-scale metabolic model (Hefzi et al. 2016). However, we found that only 23 genes were expressed above the 95% z-score expression threshold employed by gMCS (Figure 2, below). Since high expression is a necessary condition for gMCS to predict a gene as essential, such a low ratio of highly expressed genes means gMCS will underpredict essential genes. At best, only 23/451 genes in the test set could be predicted as essential, which is far below the true fraction of essential genes in the set (77/451). We suspect this limitation results from gMCS being specifically designed for predicting synthetic lethal genes in cancer cell lines, where only few double knockout pairs in the total deletion search space are essential and singly essential genes are excluded from analysis.

Figure 2: Distribution of expression levels for N=451 held-out test genes in Figure 3B of our manuscript. Median RNA-seq raw probe scores were computed from the data in (Hefzi et al, 2016) across four replicates. The 5% cutoff ($Z=1.33$) was applied to mark genes as highly expressed (dashed line).

Q: (3)The introduction lacks a comprehensive review of existing gene essentiality prediction tools, including FluxGAT, gMCS, and non-metabolic network-based methods, which weakens the claim of novelty.

A: We apologize for this oversight. We have added several additional literature sources on gene essentiality prediction in L24-31 of the Introduction.

Q: 2.Evaluation Metrics and Statistical Rigor

(1)Results are reported only in terms of accuracy and ROC-AUC. Standard ML practice requires at minimum precision, recall, and F1-score—especially where class imbalance may skew accuracy.

A: Thank you for pointing this out. In our first submission we reported the complete metrics only for the *E. coli* study (Figure 2). The new Supplementary Table S5 contains all the relevant classification metrics for *S. cerevisiae* and CHO cells.

Q: (2) No confidence intervals or statistical tests are provided to assess whether reported gains over FBA (or other methods) are significant.

A: Since FBA is deterministic, there are no confidence intervals over its predictions and for simplicity we opted to use descriptive statistics to compare with FCL. We provide standard error for all our FCL predictions in Figures 2-3 to capture variation in cross-validation as well as changes to various components of the FCL training pipeline.

Q: 3. Objective Function in Flux Sampling

(1) The authors claim FCL “does not require an objective function,” yet their sampling procedure necessarily uses the model’s default objective (typically biomass maximization). They do not explore how changing the objective—for instance to β -carotene production—would alter sampling outcomes or prediction performance.

(2) For the β -carotene classification task, it is unclear whether the flux samples were drawn under a biomass objective or a β -carotene objective. A direct comparison—sampling under each objective, and even FBA-based essentiality prediction for β -carotene—would clarify the method’s sensitivity to this choice.

A: Flux sampling does not require an objective function and it is not possible to sample the flux cone under a specific objective; see e.g. the authoritative review by Schellenberger & Palsson, *J Biol Chem*, 2009. This is the key distinction between flux sampling and FBA methods that rely on an optimality principle.

Q: 4. Thresholds and Parameter Justification

Key experimental details are missing, such as the objective function used during flux sampling for β -carotene synthesis predictions and the rationale for the growth rate threshold of 0.4 h^{-1} for gene essentiality classification. This threshold deviates from the standard 5-10% of wild-type growth rate typically used in GEM-based predictions, and its justification is unclear.

A: As mentioned, flux sampling does not use an objective function. Moreover, we did not choose a growth rate cutoff ourselves, but instead took the value (0.4 h^{-1}) directly from the original iML1515 paper (Monk et al, *Nat Biotech*, 2017) to avoid introducing additional confounders in our analysis. Figure 3 below shows the distribution of experimental growth rates for *E. coli* in glucose medium. The selected cutoff (dashed line) captures all knockouts with greater than a 50% drop in growth rate. The distribution is strongly bimodal and the majority of nonessential knockouts have less than a 10% drop in growth rate.

Likewise, in the yeast and CHO examples (Figure 3), we pulled the essentiality ground truth labels directly from previous literature for consistency of comparisons (Zhang et al., 2024 and Xiong et al 2021).

Figure 3: Growth rate distribution of experimental data used to evaluate FBA predictions of the iML1515 model for *E. coli* (Monk et al 2017). Bars are colored based on the label.

Q: 5.Code Availability

The provided code link fails (“MultCloud is not supported in this region”), preventing validation. Please host code in a publicly accessible repository (e.g. Zenodo) and include clear installation and usage instructions.

A: We apologize for this oversight. We did upload our code as a supplementary file in the submission system, but unfortunately the server we used to store the training data was not accessible from all locations.

We have now uploaded the code, usage instructions and training data for all organisms to a Zenodo repository available at DOI: [10.5281/zenodo.15518666](https://doi.org/10.5281/zenodo.15518666). The code allows reproducing all results in the manuscript.

Response to referees

Accurate prediction of gene deletion phenotypes with Flux Cone Learning

Charlotte Merzbacher, Oisín Mac Aodha, Diego A. Oyarzún

We thank reviewers for their swift response. Reviewer 1 did not raise any further concerns. Below we respond to the remaining points raised by Reviewer 2.

Q: On the Threshold for Defining Gene Essentiality (Residual Issue from Round 1):

The authors continue to define gene essentiality using a cutoff of 0.4 h^{-1} for growth rate reduction, citing Monk et al. (2017) as the source. However, the supplemental materials of the iML1515 paper (Monk et al., 2017, lines 523–524) clearly state that: "A gene was considered computationally essential for the simulated condition if deletion of the gene reduced optimal growth rate to less than 0.05 h^{-1} ."

This raises a significant concern: the 0.4 h^{-1} threshold used by the authors deviates substantially from the original 0.05 h^{-1} . This discrepancy has serious implications for the classification of essential genes, and thereby the reliability and comparability of downstream analyses.

A: We apologize for this confusion as we misread the reviewer's original query in the first round as referring to the experimental labels rather than the FBA prediction labels.

We agree that the FBA growth cutoff can impact the downstream analysis; that has been discussed at length by Bernstein et al., Mol Syst Biol, 2023, who argue that Receiver Operating Characteristic or Precision-Recall curves as more robust ways to evaluate accuracy metrics of FBA predictions.

We clarify that the superiority of FCL over FBA in *E. coli* is independent of the cutoff used for FBA predictions. This is shown by the ROC curve below, which produces the true positive and false positive rates for both methods across all possible cutoffs. The ROC curve clearly shows that FCL outperforms FBA regardless of cutoff. We have included this figure as new **Supplementary Figure 5** and updated the code repository with this analysis. The performance improvements for *S. cerevisiae* and CHO cells are likewise demonstrated by the ROC curves in Figure 3. We apologize for failing to include the ROC curve for *E. coli* in our previous submission.

Figure. Receiver operating characteristic curves for *E. coli* computed on a held-out set of test genes (20%) across five training repeats.

We have also clarified our choice of FBA growth rate cutoff employed to produce the results in Figure 2a-b (L466-471). We chose 0.4 1/hr as the cutoff to match the experimental growth rate employed by Monk et al, which is 50% of the wild-type growth rate (predicted to be ~0.8 1/hr by FBA). This can be verified in Column D of Supplementary Dataset 11 from Monk et al that contains relative growth rates (growth = 1 being equivalent to wild type). This equalized cutoff of 0.4 1/hr was used to threshold both the experimental data (matching the classifications provided in Supplementary Dataset 11) and the FBA predictions. We also note that FBA growth rate predictions are strongly bimodal and, as a result, our cutoff has minimal differences with the one suggested by the referee as per the figure below.

Figure. Distribution of FBA growth rate predictions computed from iML1515 on glucose. The suggested cutoff is in red; ours is in black. Both cutoffs produce almost identical binarized labels because of the strong bimodality of the FBA growth rate predictions.

2. On the Significance of Cross-Species Performance Improvements: The authors evaluated the performance of their method in predicting essential genes across three species: *Escherichia coli*, *Saccharomyces cerevisiae*, and Chinese Hamster Ovary (CHO) cells (Table S5). The results showed a significant improvement in *S. cerevisiae* (from 81% to 90%, a 9% increase), a modest improvement in *E. coli* (approximately 1.5%), and almost no improvement in CHO cells. Overall, the method yielded only limited performance gains across different species.

A: While for *E. coli*, the gains are seemingly small, we note that the margins for improvement are already quite tight because both FBA and FCL are extremely accurate. For example, an accuracy on essential genes of 76% (FBA) and 82% (FCL) means that the algorithms can correctly call 39/50 and 41/50 of essential genes in the test set, respectively.

We agree that the performance improvement for CHO cells is narrow. As we mention in the paper, it is plausible that this results from incompleteness of the CHO genome-scale model, and we have expanded on this argument in the discussion of the CHO model results (L138-142). This is also supported by the observation that FBA predictions for CHO cells are generally less accurate than other model organisms with better curated genome-scale metabolic reconstructions, as recently discussed by Strain et al, *Biotechnology and Bioengineering*, 2023.

Q: On the Practical Utility and Validation of Small Molecule Synthesis Prediction. Figure 4B shows that SVG, LogReg, RF methods achieve accuracy close to or below 50%, essentially equivalent to random guessing. The HGB method HGB achieves ~63–67% accuracy (as visually estimated, though the manuscript reports 69.8% — please confirm consistency with the figure). In biological design or metabolic engineering, is this level of accuracy (~70%) sufficient to guide decision-making? A deeper discussion on the practical significance and limitations of this result is warranted. If possible, the authors are strongly encouraged to select a few representative predictions and carry out wet-lab validation experiments. Even a limited number of validations would significantly strengthen the manuscript's impact and demonstrate real-world applicability.

A: In Figure 4 we reported results across a wide range of models, some of which performed better than others depending on whether we focus on overall accuracy or per-class accuracy. For example, the LogReg and SVC classifiers get low overall accuracy but higher high-producer accuracy (Supplementary Table S6).

We agree that accuracies are not stellar. This results from the production data being strongly skewed toward mid-producers, and high producers are underrepresented (see histogram in Figure 4a). This data bias is likely to appear for many small molecules and production hosts, simply because boosting production is challenging, and few knockouts can improve production phenotypes. Since FCL is the first and only method able to predict small molecule synthesis from knockout screens, we expect the machine learning community to quickly pick up the foundations laid by FCL and develop model architectures or data augmentation strategies for improved performance. We remain excited about these future developments.

The reviewer raises an important point: what is an acceptable accuracy for forward design in metabolic engineering? This is a cross-cutting question across many applications of machine

learning for synthetic biology, for which there is no universal answer. We have previously addressed this issue extensively in the context of sequence-to-expression modelling (Nikolados et al, Nat Commum, 2022; Nikolados et al, Curr Op Biotech, 2023). For applications of FCL in metabolic engineering, model accuracy needs to be relativized to the experimental reproducibility of production readouts, cost of implementation, and risk of failure. Experimental reproducibility puts a ceiling on what we should expect from a machine learning model, as otherwise the models would be predicting with higher accuracy than the measurement error. Cost and risk of failure, in turn, determine how lenient we can be on model predictions. We have added an additional discussion on this topic to the paper (L217-227).

With regards to wetlab validation, we highlight that all performance metrics were computed on knockouts not seen in training, and therefore they can be seen as post-hoc experimental validation. Fresh wetlab validation would require using FCL to discover and validate new yeast knockouts strains with improved betaxanthin production. We feel this lies beyond the remit of our manuscript. After discussing with the Editor, they agreed to waive this requirement given the scope of our paper and our logistical constraints.

Response to referees

Accurate prediction of gene deletion phenotypes with Flux Cone Learning

Charlotte Merzbacher, Oisín Mac Aodha, Diego A. Oyarzún

We thank the remaining reviewer for their swift response and positive comments about our manuscript. As agreed with the editor, there are no outstanding concerns with our manuscript.